# Impact of Breastfeeding Duration on Adenoid Hypertrophy, Snoring and Acute Otitis Media: A Case-Control Study in Preschool Children

**DOI:** 10.3390/jcm12247683

**Published:** 2023-12-14

**Authors:** Aleksander Zwierz, Krzysztof Domagalski, Krystyna Masna, Paweł Walentowicz, Paweł Burduk

**Affiliations:** 1Department of Otolaryngology, Phoniatrics and Audiology, Faculty of Health Sciences, Ludwik Rydygier Collegium Medicum, Nicolaus Copernicus University, 75 Ujejskiego Street, 85-168 Bydgoszcz, Poland; krymasna@gmail.com (K.M.); pburduk@wp.pl (P.B.); 2Department of Immunology, Faculty of Biological and Veterinary Sciences, Nicolaus Copenicus University, 87-100 Toruń, Poland; krydom@umk.pl; 3Department of Obstetrics, Gynecology and Gynecologic Oncology, Regional Polyclinical Hospital, 87-100 Toruń, Poland; walentowiczp@gmail.com

**Keywords:** adenoid hypertrophy, breastfeeding, snoring, open mouth breathing, acute otitis media

## Abstract

Background: The aim of this study was to analyze the relationship between breastfeeding duration and adenoid size, snoring and acute otitis media (AOM). Methods: We analyzed the medical history of children admitted to the ENT outpatient clinic in 2022 and 2023, reported symptoms, ear, nose and throat (ENT) examination, and flexible nasopharyngoscopy examination of 145 children aged 3–5 years. Results: Breastfeeding duration of 3 and 6 months or more had a significant effect on the reduction of snoring (*p* = 0.021; *p* = 0.039). However, it had no effect on the adenoid size, mucus coverage and sleeping with an open mouth. Snoring was correlated with open mouth sleeping (*p* < 0.001), adenoid size with a 75% A/C ratio or more (*p* < 0.001), and adenoid mucus coverage in the Mucus of Adenoid Scale by Nasopharyngoscopy Assessment—MASNA scale (*p* = 0.009). Children who were breastfed for less than 3 months had more than a four-fold greater risk of snoring. There was a statistically significant correlation between AOM and gender (*p* = 0.033), breastfeeding duration in groups fed 1, 3 or 6 months or more (*p* = 0.018; *p* = 0.004; *p* = 0.004) and those fed with mother’s breast milk 3 or 6 months or more (*p* = 0.009; *p* = 0.010). Moreover, a correlation was found between adenoid size and mucus coverage, tympanogram, and open-mouth sleeping (*p* < 0.001). Independent factors of snoring in 3- to 5-year-old children were breastfeeding duration of less than 3 months (*p* = 0.032), adenoid size with an A/C ratio of 75% or more (*p* = 0.023) and open mouth sleeping (*p* = 0.001). Conclusions: Children breastfed for 3 and 6 months or more exhibited reduced rates of snoring. There was no effect of breastfeeding duration on adenoid size in children aged 3 to 5 years, suggesting that the link between breastfeeding duration and snoring is primarily associated with craniofacial development and muscle tone stimulation. A breastfeeding duration of 1 month or more plays a key role in reducing the rate of AOM. The mother’s milk plays a protective role against AOM. The presence of mucus might be responsible for snoring in preschool children. A medical history of breastfeeding should be taken into consideration when snoring children are suspected of adenoid hypertrophy.

## 1. Introduction

The World Health Organization (WHO) strongly advocates breastfeeding for infants, recommending it from birth until at least six months due to its remarkable impact on bolstering the infant’s immune system and promoting optimal craniofacial development [1,2]. Furthermore, the United Nations International Children’s Emergency Fund (UNICEF) recommends the introduction of complementary foods alongside breastfeeding from 6 to 24 months [1].

The quality and composition of breast milk exhibit variations depending on the postpartum period. Colostrum, produced from birth to the 7th day, transitions into transitional milk from the 7th to the 15th day, ultimately giving way to mature milk four weeks postpartum [3,4]. Initially abundant in immune components, colostrum undergoes a decrease, stabilizing as breast milk matures [5]. Simultaneously, the milk’s composition, inclusive of minerals, vitamins, hormones, enzymes, nucleic acids, polysaccharides, lipids, breast-derived cells, blood-derived cells, and extracellular vesicles, undergoes alterations [4,5,6]. Consequently, the duration of breastfeeding can impact a child’s susceptibility to infectious diseases, allergies, or immune disorders in the future [7].

Breast milk encompasses various bioactive components, with a pivotal role attributed to sIgA immunoglobulins. These immunoglobulins safeguard the infant’s intestinal barrier, reduce the risk of upper and lower respiratory tract infections, and mitigate inflammation [8]. This immunoprotective effect is believed to be mediated through the passive transfer of maternal immune components during breastfeeding. Furthermore, breast milk is instrumental in shaping the upper respiratory tract microbiome and fostering greater alpha diversity [9]. Breast-derived microbiota (BDM) contribute to the development of offspring microbiota through mucosal colonization, influencing the maturation of the immune system [10]. Notably, significant variations in the microbiome exist between infants fed mother’s milk and those fed formula [11,12].

Breastfeeding contributes significantly to the proper development of the oropharyngeal apparatus, including the peristaltic movement of the tongue during suckling, which in turn aids in the development and coordination of the oropharyngeal muscles engaged in swallowing [13]. Additionally, it diminishes the risk of malocclusion disorders in children and supports the healthy formation of the palate [14,15,16]. Importantly, scientific evidence has underscored a connection between the duration of breastfeeding and the subsequent occurrences of habitual snoring and apnea in preschool and early school-aged children. Habitual snoring is observed in 9% of preschool children, with adenoid hypertrophy being the predominant causative factor [17,18]. The aim of this study is to pursue and elucidate the possible association between the duration of breastfeeding, snoring, adenoid size and acute otitis media (AOM). Specifically, it explores whether sustained exposure to immune components in breast milk has an influence on adenoid growth and mucus coverage, or if the link between breastfeeding duration and snoring is primarily attributable to craniofacial development and muscle tone stimulation.

## 2. Materials and Methods

*Study Population:* The study encompassed 145 children (60 girls and 85 boys) aged 3–5 years, presenting with symptoms suggestive of adenoid hypertrophy, who were evaluated at an ear, nose, and throat (ENT) outpatient clinic in 2022 and 2023. Detailed assessments included allergy history, symptoms linked to adenoid hypertrophy (e.g., snoring, mouth breathing), stated by parents’ hypoacusis, frequency of upper respiratory tract infections, and the presence and severity of a runny nose. Snoring was defined in accordance with criteria based on the International Classification of Sleep Disorders (ICSD-3) [19]. The children were categorized into six groups based on breastfeeding duration: up to 3 month, 3 to 6 months, 6 to 12 months, 12 to 18 months, 18 to 24 months, and over 24 months. To determine the critical duration of breastfeeding, various analyses were conducted on the entire cohort of children, involving the subdivision into two subgroups with variable cut-off points: below and above 1 month, up to 3 months and over 3 months, up to 6 months and over 6 months, up to 12 months and over 12 months, up to 24 months and over 24 months.

Additionally, we analyzed children according to the same categorized periods of mother’s milk feeding, both from the breast and the bottle. The groups were analyzed according to adenoid size, mucus coverage and related symptoms, such as snoring and sleeping with an open mouth. Additionally, a history of AOM and the results of tympanometry examinations were recorded and compared with the breastfeeding period. Exclusion criteria included recent and prolonged treatment for adenoid hypertrophy with nasal steroid spray, craniofacial anomalies, genetic disorders (e.g., Down syndrome), nasal septal deviation, nasal polyps, inferior turbinate hypertrophy, and active upper respiratory infections. The patient characteristics and reported symptoms are presented in Table 1.

*Endoscopy:* Flexible endoscopic examinations of the nasopharynx were conducted on each child using a common nasal meatus. These examinations were carried out by a pediatric otorhinolaryngologist (AZ) using the Karl Storz Tele Pack endoscopic system, which was equipped with a flexible nasopharyngoscopy tool (2.8 mm outer diameter, 300 mm length). The assessment included choanal obstruction percentages (adenoid-to-choanae ratio—A/C ratio) and mucus coverage of the adenoids based on flexible nasopharyngoscopy. Choanal obstructions were assessed with up to 5% accuracy. To assess adenoid size in this study, we divided children into two groups: those with an adenoid size less than 75% in the A/C ratio, and those with 75% or more in the A/C ratio in the flexible nasopharyngoscopy assessment. This categorization was derived from our intraoperative comparisons of adenoid size with preoperative endoscopic adenoid assessments, indicating that a 75% A/C ratio or more was equivalent to a surgically removed large adenoid [20], which was also confirmed in other clinicians’ opinions and studies [21,22]. Videos of the nasopharynx were coded and analyzed blindly by the second ENT doctor (KM). Furthermore, the Mucus of Adenoid Scale by Nasopharyngoscopy Assessment (MASNA) was utilized to quantify adenoid mucus coverage, employing a four-point scale (0, no mucus; 1, residue of clear watery mucus; 2, some amount of dense mucus; 3, copious thick dense mucus) [23]. In cases of assessment discrepancies, a third ENT doctor (PB) reassessed the score.

*Tympanometry:* The study involved otoscopic examinations and external auditory canal cleaning when required. Tympanometry was carried out using the GSI 39 AutoTymp TM by Grason-Stadler. Effusion in the middle ear was assessed through tympanometry measurements and tympanogram graphs. The results were classified using the Liden and Jerger classification system for tympanograms [24,25]. Tympanograms were saved for each patient’s right and left ears, but to simplify the score, we divided them into three groups in accordance with the worst tympanogram in both ears. We adopted type-B tympanograms to consider the worst, type-C worst, and type-A indicative of normal function.

*Statistical Analysis:* Descriptive statistics were used to summarize variables within the study group. Quantitative variables were presented as means ± standard deviation (SD), while categorical variables were summarized using frequency counts and percentages. Statistical significance was determined using the Chi-square method or Fisher’s exact test for categorical variables and Student’s *t*-test or one-way ANOVA for quantitative variables to assess differences between independent variables.

Variables significantly related to snoring in the univariate analysis were included in the logistic regression analysis to identify independent prognostic factors useful in assessing snoring. Prediction of snoring was assessed via two separate analyses: using (1) breastfeeding < 3 months and (2) breastfeeding < 6 months for the logistic regression models. Odds ratios (OR) and 95% confidence intervals (95% CIs) were also calculated for considered clinical variables in regression models. For all these tests, two-tailed *p*-values were used, and differences at the level of *p* < 0.05 were considered significant. All statistical analyses were performed with SPSS (Statistical Package for the Social Sciences, version 28, Armonk, NY, USA) software.

*Ethics:* Ethical approval for this study was obtained from the ethics committee of Nicolaus Copernicus University (KB 141/2022).

## 3. Results

The study comprised 145 children with an average age of 3.9 years, a mean breastfeeding duration of 13 months, and an average adenoid size corresponding to a 65.3% A/C ratio (Table 1). The duration of breastfeeding 6 months or more had a significant effect on the reduction of snoring and incidence of AOM (*p* = 0.039 and *p* = 0.004, respectively). However, it had no effect on the adenoid size, mucus coverage, nasal blockage, sleeping with an open mouth, frequency of recurrent upper respiratory tract infection, rhinitis, cough, hypoacusis and tympanogram (Table 1).

To assess the factors that impact snoring within the study sample, a comparison was made between children who snored and those who did not (Table 2). There was no significant difference between age, gender, allergy history, hypoacusis, recurrent upper respiratory tract infections, rhinitis, cough, or nasal blockage. However, a statistically significant difference in snoring rates was observed based on breastfeeding duration. Children breastfed for 3 months, 6 months or more exhibited reduced rates of snoring (*p* = 0.021 and *p* = 0.039, respectively). Also, children fed with mother’s milk for 3 months or more snored less (*p* = 0.014). Snoring was also correlated with open mouth sleeping (*p* < 0.001), adenoid size with an A/C ratio of 75% or more (*p* < 0.001), and adenoid mucus coverage with the presence of any mucus (MASNA 1 to 3 degrees) (*p* = 0.009) (Table 2). Children with an adenoid size with an A/C ratio of 75% or more and mucus coverage on the adenoid snore more often. No relationship was observed between snoring and allergy, AOM, recurrent respiratory tract infections, rhinitis, nose blockage, cough, hypoacusis and tympanogram.

Further relationships between the history of AOM and other factors were analyzed (Table 2). We found a statistically significant correlation between AOM and gender (*p* = 0.033). The girls became sick more often. Breastfeeding duration, especially in groups fed 1, 3 or 6 months and more, impacted the recurrence of AOM (*p* = 0.018, *p* = 0.004 and *p* = 0.004, respectively). One month of breastfeeding was enough to reduce the amount of AOM. Also, in the groups of children fed with breast milk for 3 months, 6 months or more, a statistically significant difference in the incidence of AOM was shown (*p* = 0.009 and *p* = 0.010, respectively). A high correlation was also found depending on the depending on the type A tympanogram (*p* = 0.003). Tympanometry, indicating any presence of fluid in the middle ear, fostered the occurrence of AOM. There was no stated correlation between the history of AOM and adenoid size, adenoid mucus coverage, rhinitis, or upper respiratory tract infection, as observed by parents’ hypoacusis, cough, or snoring.

In the next step, correlations between the adenoid size and previously analyzed clinical data were analyzed. In addition to the previously presented correlations between adenoid size and snoring and the lack of correlation with breastfeeding duration, a statistically significant correlation was found between adenoid size and mucus coverage, tympanogram or open mouth sleeping (Table 3). In all of these correlation cases, the calculated *p*-value was <0.001. Adenoid hypertrophy of 75% and more was conducive to greater adenoid mucus coverage, C or B tympanogram, and sleeping with an open mouth. Additionally, a relationship was detected between adenoid size and mucus coverage on the MASNA scale, whereas larger adenoids more frequently displayed pathological mucus coverage. The presence of effusion in the middle ear was also linked to bigger adenoid size (Table 3). Moreover, it was determined that adenoid mucus coverage influenced snoring and tympanometry results.

Finally, we assessed the importance of breastfeeding in the prediction of snoring using logistic regression analysis. Regression analysis model using a 3-month breastfeeding cut-off point showed that a breastfeeding duration of less than 3 months (*p* = 0.032), adenoid size with an A/C ratio of 75% or more (*p* = 0.023) and open mouth sleeping (*p* = 0.001) were independent factors of snoring in children 3 to 5 years old (Table 4). The obtained estimates indicate that children who were breastfed for less than 3 months had more than a 4-fold greater risk of snoring than those who were breastfed for more than 3 months (OR = 4.33, 95% CI = 1.14–16.50). Analyzing the model with a 6-month breastfeeding cut-off point, sleeping with an open mouth (*p* < 0.001) and adenoid size of 75% or more (*p* = 0.030) were shown as significant factors of snoring.

## 4. Discussion

The study did not show an effect of breastfeeding duration on the adenoid size in children aged 3–5 years. No other studies in the available literature database have analyzed this relationship.

However, the study affirmed the beneficial effect of breastfeeding for 3 and 6 months or more in reducing snoring rates among children aged 3 to 5 years. This finding aligns with previous studies by Brew et al. indicating that breastfeeding for over one month is associated with a reduced risk of parent-reported habitual snoring [26]. Moreover, breastfeeding for longer than 3 months was linked to a significant decrease in the risk of witnessed sleep apnea. Beebe et al. and Montgomery-Downs et al. also support this observation, suggesting that breastfeeding may confer protection against sleep-disordered breathing [17,27]. One proposed hypothesis centers on the immunological protection of maternal milk, which reduces early childhood infections and inhibits lymphoid tissue proliferation. At ages 3 to 5, adenoids represent the most developed lymphoid tissue within Waldeyer’s ring [20]. However, this study did not validate a correlation between breastfeeding duration and adenoid size as measured via flexible nasofiberscopy. Moreover, no such relationship emerged when considering the duration of breastfeeding. Multivariate analysis indicated that breastfeeding duration and adenoid size independently contributed to snoring. This raises questions regarding the second hypothesis postulated by Montgomery-Downs, suggesting that the link between breastfeeding duration and snoring might be primarily associated with craniofacial development and muscle tone stimulation [27,28].

Our study has shown the beneficial effects of breastfeeding for more than one month on reducing the risk of AOM in preschool children. This was already reported in the literature and meta-analysis performed by other authors [29,30,31,32,33]. The population studies in Canada by Karunanayake et al. indicated that breastfeeding for more than 3 months protects against ear infection. We have also shown a beneficial effect of feeding the baby with breast milk for more than 3 months on reducing the incidence of AOM. This confirms the suggested effect of immune modulation, epigenetic changes, and colonization of beneficial microbiota supplemented with mother’s milk [32,34]. Moreover, mothers’ milk has a bacteriostatic effect and contains lactoperoxidase, which destroys pathogenic AOM bacteria [32]. The positive effect of a short breastfeeding period (one month in our study) on the reduction of the incidence of AOM supports the protective role of mother’s milk against AOM rather than the breastfeeding process itself, influencing the development of muscle tone of the oropharyngeal apparatus and improving Eustachian tube function. Perhaps this period is sufficient to shape baby’s immunological system by mother’s milk, which is then exceptionally rich in immune components [5].

The incidence of AOM correlated with tympanometry results, indicating effusion in the middle ear confirmed by a type C or B tympanogram. This clearly confirmed that effusion in the middle ear promotes the growth of pathogens and increases the risk of otitis media [35].

One noteworthy revelation in this study was the identification of a potential link between adenoid mucus coverage and snoring in preschool children. The presence of mucus is responsible for constriction of the nose and nasopharynx, increasing upper airway resistance and influencing snoring. Consequently, snoring in preschool children may be misinterpreted as a symptom of adenoid hypertrophy, even though it can be substantially caused by mucus covering the adenoid. Nonetheless, appropriate treatments such as anti-allergic medications, nasal steroids, and nasal lavage can potentially mitigate this effect. Research by Masna et al. emphasized that other factors, such as the thermal season, can also impact adenoid mucus [23]. This observation was also stated by Wang et al., who showed seasonal dependence of snoring problems related to nasal blockage in the general population [36]. A reduction in nasal resistance plays an important role in reducing snoring [37]. This was corroborated by Värendh et al., who identified nasal symptoms as independent risk factors for snoring [38].

Our study did not show a relationship between the length of breastfeeding and open mouth sleeping. This is opposite to the results of a meta-analysis conducted by Savian et al., indicating a possible protective effect against the occurrence of mouth breathing [39]. A study by Lopes et al. indicated that an increased duration of breastfeeding increases the likelihood of developing a normal breathing pattern in 2.5- to 4-year-old children [40]. Similar positive results of breastfeeding for more than 3 months in similar groups of preschool children were shown by Travitzki [41]. Similar to our findings, Ieto did not find that feeding correlated with predominant breathing patterns [42].

In addition, we confirmed the reports that other authors have already shown regarding the relationship between adenoid hypertrophy and the presence of effusion in the middle ear confirmed by tympanometry examination or adenoid size and sleeping with the mouth open [43,44,45,46,47]. The high convergence of the obtained results and the resulting correlations with previous studies by other authors indicate that the studied group is representative. This increases the credibility of our results, particularly the lack of correlation between breastfeeding periods and adenoid hypertrophy.

Finally, it has been shown that the breastfeeding period and adenoid hypertrophy are independent factors influencing the snoring of preschool children. Therefore, a snoring child breastfed under 3 months may be misdiagnosed due to adenoid hypertrophy. A medical history of breastfeeding should be taken into consideration when snoring children are suspected of adenoid hypertrophy.

The study’s robustness lies in the considerable size of the studied children’s group, substantial congruence with previous studies by other researchers, and a pioneering attempt to explore the correlation between breastfeeding duration and adenoid size assessed with optimal objectivity, utilizing flexible endoscopy. It is plausible that further analyses on larger patient cohorts would afford a clearer determination of the significance level for the observed relationships, particularly in cases where *p*-values hover near 0.05.

## 5. Conclusions

This study underscored the positive impact of breastfeeding for a minimum of 3 months in reducing snoring rates and preventing open month breathing among children aged 3 to 5 years. However, it did not establish any effect of breastfeeding duration on adenoid size. This reinforces the hypothesis that the favorable effect of breastfeeding on snoring is related to its influence on proper facial skeletal development, occlusion, and muscle tension in the mouth and throat. Moreover, it was found that breastfeeding duration of 3 months or more and adenoid hypertrophy with a 75% A/C ratio or more are independent factors of snoring in preschool children. Breastfeeding duration should be taken into account when, based on symptoms such as snoring, the doctor makes a diagnosis of suspected adenoid hypertrophy. The presence of mucus might be responsible for snoring in preschool children. Breastfeeding duration of 1 month or more plays a key role in reducing the rate of AOM. The nutrients of a mother’s milk play a protective role against AOM.

## Figures and Tables

**Table 1 jcm-12-07683-t001:** Characteristics of the study group divided into a 6-month breastfeeding period.

Characteristic		All Patients (*n* = 145)	Breastfeeding Groups		*p* Value
			≥6 Months (*n* = 100)	<6 Months (*n* = 45)	
Age (years)	mean ± SD	3.9 ± 0.8	3.9 ± 0.8	3.9 ± 0.8	0.981
Gender	female	60 (41.4%)	41 (41.0%)	19 (42.2%)	0.890
	male	85 (58.6%)	59 (59.0%)	26 (57.8%)	
Breastfeeding (months)	mean ± SD	13.0 ± 11.2			
	0–3	26 (17.9%)	-	-	-
	3–6	14 (9.7%)			
	6–12	31 (21.4%)			
	12–18	21 (14.5%)			
	18–24	16 (11.0%)			
	>24	37 (25.5%)			
Adenoid size (A/C ratio, %)	mean ± SD	65.3 ± 18.8	64.4 ± 19.0	67.3 ± 18.5	
	<75	90 (62.1%)	66 (66.0%)	24 (53.3%)	0.146
	≥75	55 (37.9%)	34 (34.0%)	21 (46.7%)	
Adenoid mucus coverage (MASNA scale)	0	40 (27.6%)	27 (27.0%)	13 (28.9%)	0.615
	1	36 (24.8%)	28 (28.0%)	8 (17.8%)	
	2	40 (27.6%)	26 (26.0%)	14 (31.1%)	
	3	29 (20.0%)	19 (19.0%)	10 (22.2%)	
	0	40 (27.6%)	27 (27.0%)	13 (28.9%)	0.814
	1–3	105 (72.4%)	73 (73.0%)	32 (71.1%)	
Tympanogram	AA	65 (44.8%)	48 (48.0%)	17 (37.8%)	0.602
	AB/BA	6 (4.1%)	5 (5.0%)	1 (2.2%)	
	AC/CA	15 (10.3%)	10 (10.0%)	5 (11.1%)	
	BB	34 (23.4%)	23 (23.0%)	11 (24.4%)	
	BC/CB	6 (4.1%)	4 (4.0%)	2 (4.4%)	
	CC	19 (13.1%)	10 (10.0%)	9 (20.0%)	
	A	65 (44.8%)	48 (48.0%)	17 (37.8%)	0.306
	B	46 (31.7%)	32 (32.0%)	14 (31.1%)	
	C	34 (23.4%)	20 (20.0%)	14 (31.1%)	
Allergy	yes	24 (16.6%)	14 (14.0%)	10 (22.2%)	0.137
	no	25 (17.2%)	21 (21.0%)	4 (8.9%)	
	not tested	96 (66.2%)	65 (65.0%)	31 (68.9%)	
Snoring	yes	106 (73.1%)	68 (68.0%)	38 (84.4%)	**0.039**
	no	39 (26.9%)	32 (32.0%)	7 (15.6%)	
Sleeping with an open mouth	yes	72 (49.7%)	53 (53.0%)	19 (42.2%)	0.394
	periodic	50 (34.5%)	31 (31.0%)	19 (42.2%)	
	no	23 (15.9%)	16 (16.0%)	7 (15.6%)	
Hypoacusis (stated by parents)	yes	35 (24.3%)	25 (25.3%)	10 (22.2%)	0.579
	periodic	25 (17.4%)	15 (15.2%)	10 (22.2%)	
	no	84 (58.3%)	59 (59.6%)	25 (55.6%)	
History of acute otitis media	yes	84 (57.9)	50 (50.0%)	11 (24.4%)	**0.004**
	no	61 (42.1%)	50 (50.0%)	34 (75.6%)	
rURTIs	yes	120 (82.8%)	83 (83.0%)	37 (82.2%)	0.909
	no	25 (17.2%)	17 (17.0%)	8 (17.8%)	
Rhinitis (weeks in a month)	mean ± SD	1.8 ± 1.0	1.9 ± 1.0	1.6 ± 0.9	0.072
Cough	yes	48 (33.1%)	37 (37.0%)	11 (24.4%)	0.237
	no	97 (66.9%)	63 (63.0%)	34 (75.6%)	
Blocked nose	yes	11 (7.6%)	7 (7.0%)	4 (8.9%)	0.739
	no	134 (92.4%)	93 (93.0%)	41 (91.1%)	

A/C ratio: adenoid-to-choana ratio, rURTIs: recurrent upper respiratory tract infections.

**Table 2 jcm-12-07683-t002:** The relationship of snoring and the occurrence of acute otitis media with clinical and demographic data in children aged 3–5 years with suspected adenoid hypertrophy.

Characteristic		History of Acute Otitis Media		*p* Value	Snoring		*p* Value
		Yes (*n* = 84)	No (*n* = 61)		Yes (*n* = 106)	No (*n* = 39)	
Age (years)	mean ± SD	4.0 ± 0.8	3.8 ± 0.9	0.345	4.0 ± 0.8	3.8 ± 0.8	0.211
Gender	female	41 (48.8%)	19 (31.1%)	**0.033**	46 (43.4%)	14 (35.9%)	0.416
	male	43 (51.2%)	42 (68.9%)		60 (56.6%)	25 (64.1%)	
Breastfeeding (months)	mean ± SD	11.8 ± 11.5	14.8 ± 10.5	0.107	12.5 ± 11.3	14.6 ± 10.7	0.318
	≥1	66 (78.6%)	57 (93.4%)	**0.018**	87 (82.1%)	36 (92.3%)	0.128
	<1	18 (21.4%)	4 (6.6%)		19 (17.9%)	3 (7.7%)	
	≥3	59 (70.2%)	55 (90.2%)	**0.004**	78 (73.6%)	36 (92.3%)	**0.021**
	<3	25 (29.8%)	6 (9.8%)		28 (26.4%)	3 (7.7%)	
	≥6	50 (59.5%)	50 (82.0%)	**0.004**	68 (64.2%)	32 (82.1%)	**0.039**
	<6	34 (40.5%)	11 (18.0%)		38 (35.8%)	7 (17.9%)	
	≥12	38 (45.2%)	33 (54.1%)	0.292	49 (46.2%)	22 (56.4%)	0.277
	<12	46 (54.8%)	28 (45.9%)		57 (53.8%)	17 (43.6%)	
	≥24	18 (21.4%)	19 (31.1%)	0.185	26 (24.5%)	11 (28.2%)	0.652
	<24	66 (78.6%)	42 (68.9%)		80 (75.5%)	28 (71.8%)	
Mother’s milk feeding	mean ± SD	12.3 ± 11.2	15.0 ± 10.3	0.139	12.9 ± 11.0	14.9 ± 10.5	0.329
	≥1	71 (84.5%)	58 (95.1%)	0.060	87 (82.1%)	36 (92.3%)	0.236
	<1	13 (15.5%)	3 (4.9%)		19 (17.9%)	3 (7.7%)	
	≥3	63 (75.0%)	56 (91.8%)	**0.009**	78 (73.6%)	36 (92.3%)	**0.014**
	<3	21 (25.0%)	5 (8.2%)		28 (26.4%)	3 (7.7%)	
	≥6	54 (64.3%)	51 (83.6%)	**0.010**	68 (64.2%)	32 (82.1%)	0.059
	<6	30 (35.7%)	10 (16.4%)		38 (35.8%)	7 (17.9%)	
	≥12	40 (47.6%)	34 (55.7%)	0.334	49 (46.2%)	22 (56.4%)	0.246
	<12	44 (52.4%)	27 (44.3%)		57 (53.8%)	17 (43.6%)	
	≥24	18 (21.4%)	19 (31.1%)	0.185	26 (24.5%)	11 (28.2%)	0.652
	<24	66 (78.6%)	42 (68.9%)		80 (75.5%)	28 (71.8%)	
Adenoid size (A/C ratio, %)	mean ± SD	66.6 ± 19.0	63.4 ± 18.6	0.320	68.2 ± 18.7	57.4 ± 17.2	**0.002**
	<75	47 (56.0%)	43 (70.5%)	0.075	56 (52.8%)	34 (87.2%)	**<0.001**
	≥75	37 (44.0%)	18 (29.5%)		50 (47.2%)	5 (12.8%)	
Adenoid mucus coverage (MASNA scale)	0	22 (26.2%)	18 (29.5%)	0.888	23 (21.7%)	17 (43.6%)	0.060
	1	22 (26.2%)	14 (23.0%)		29 (27.4%)	7 (17.9%)	
	2	22 (26.2%)	18 (29.5%)		30 (28.3%)	10 (25.6%)	
	3	18 (21.4%)	11 (18.0%)		24 (22.6%)	5 (12.8%)	
	0	22 (26.2%)	18 (29.5%)	0.659	23 (21.7%)	17 (43.6%)	**0.009**
	1–3	62 (73.8%)	43 (70.5%)		83 (78.3%)	22 (56.4%)	
Tympanogram	AA	29 (34.5%)	36 (59.0%)	0.054	50 (47.2%)	15 (38.5%)	0.486
	AB/BA	3 (3.6%)	3 (4.9%)		3 (2.8%)	3 (7.7%)	
	AC/CA	9 (10.7%)	6 (9.8%)		11 (10.4%)	4 (10.3%)	
	BB	25 (29.8%)	9 (14.8%)		26 (24.5%)	8 (20.5%)	
	BC/CB	5 (6.0%)	1 (1.6%)		3 (2.8%)	3 (7.7%)	
	CC	13 (15.5%)	6 (9.8%)		13 (12.3%)	6 (15.4%)	
	A	29 (34.5%)	36 (59.0%)	**0.011**	50 (47.2%)	15 (38.5%)	0.643
	B	33 (39.3%)	13 (21.3%)		32 (30.2%)	14 (35.9%)	
	C	22 (26.2%)	12 (19.7%)		24 (22.6%)	10 (25.6%)	
Allergy	yes	15 (17.9%)	9 (14.8%)	0.295	17 (16.0%)	7 (17.9%)	0.402
	no	11 (13.1%)	14 (23.0%)		21 (19.8%)	4 (10.3%)	
	not tested	58 (69.0%)	38 (62.3%)		68 (64.2%)	28 (71.8%)	
Sleeping with an open mouth	yes	44 (52.4%)	28 (45.9%)	0.709	61 (57.5%)	11 (28.2%)	**<0.001**
	periodic	28 (33.3%)	22 (36.1%)		37 (34.9%)	13 (33.3%)	
	no	12 (14.3%)	11 (18.0%)		8 (7.5%)	15 (38.5%)	
Hypoacusis (stated by parents)	yes	24 (28.9%)	11 (18.0%)	0.315	26 (24.8%)	9 (23.1%)	0.830
	periodic	14 (16.9%)	11 (18.0%)		17 (16.2%)	8 (20.5%)	
	no	45 (54.2%)	39 (63.9%)		62 (59.0%)	22 (56.4%)	
Snoring	yes	65 (77.4%)	41 (67.2%)	0.173	-	-	-
	no	19 (22.6%)	20 (32.8%)		-	-	
History of acute otitis media	yes	-	-	-	41 (38.7%)	20 (51.3%)	0.173
	no	-	-		65 (61.3%)	19 (48.7%)	
rURTIs	yes	72 (85.7%)	48 (78.7%)	0.269	91 (85.8%)	29 (74.4%)	0.104
	no	12 (14.3%)	13 (21.3%)		15 (14.2%)	10 (25.6%)	
Rhinitis (weeks in a month)	mean ± SD	1.8 ± 0.9	1.8 ± 1.1	0.711	1.9 ± 0.9	1.6 ± 1.1	0.176
Cough	yes	30 (35.7%)	18 (29.5%)	0.433	35 (33.0%)	13 (33.3%)	0.972
	no	54 (64.3%)	43 (70.5%)		71 (67.0%)	26 (66.7%)	
Blocked nose	yes	7 (8.3%)	4 (6.6%)	0.761	9 (8.5%)	2 (5.1%)	0.498
	no	77 (91.7%)	57 (93.4%)		97 (91.5%)	37 (94.9%)	

**Table 3 jcm-12-07683-t003:** Relationship of adenoid size and demographic or clinical factors in children aged 3–5 years with suspected adenoid hypertrophy.

Characteristic		Adenoid Size (A/C Ratio)		*p* Value
		n	Mean ± SD	
Gender	female	60	67.5 ± 18.0	0.234
	male	85	63.7 ± 19.4	
Breastfeeding (months)	0–3	31	66.0 ± 21.1	0.753
	3–6	14	70.4 ± 10.6	
	6–12	29	61.6 ± 20.0	
	12–18	18	63.1 ± 20.4	
	18–24	16	67.5 ± 17.3	
	>24	37	65.8 ± 18.8	
Mother’s milk feeding	0–3	26	66.5 ± 19.4	0.742
	3–6	14	70.4 ± 10.6	
	6–12	31	62.1 ± 19.6	
	12–18	21	62.4 ± 23.1	
	18–24	16	67.5 ± 17.3	
	>24	37	65.8 ± 18.8	
Adenoid mucus coverage (MASNA scale)	0	40	53.5 ± 19.9	**<0.001**
	1	36	65.3 ± 19.6	
	2	40	70.1 ± 14.5	
	3	29	74.8 ± 13.3	
	0	40	53.5 ± 19.9	**<0.001**
	1–3	105	69.8 ± 16.4	
Tympanogram	AA	65	58.8 ± 20.6	**0.010**
	AB/BA	6	75.8 ± 12.8	
	AC/CA	15	67.7 ± 13.7	
	BB	34	70.1 ± 18.7	
	BC/CB	6	72.5 ± 12.1	
	CC	19	71.1 ± 13.0	
	A	65	58.8 ± 20.6	**<0.001**
	B	46	71.2 ± 17.2	
	C	34	69.6 ± 13.2	
Allergy	yes	24	64.4 ± 19.3	0.904
	no	25	64.2 ± 17.7	
	not tested	96	65.8 ± 19.2	
Sleeping with an open mouth	yes	72	71.9 ± 15.7	**<0.001**
	periodic	50	60.2 ± 19.8	
	no	23	55.4 ± 18.8	
Hypoacusis (stated by parents)	yes	35	69.1 ± 18.0	0.283
	periodic	25	61.8 ± 22.8	
	no	84	64.3 ± 17.7	
Snoring	yes	106	68.2 ± 18.7	**0.002**
	no	39	57.4 ± 17.2	
History of acute otitis media	yes	84	66.6 ± 19.0	0.320
	no	61	63.4 ± 18.6	
rURTIs	yes	120	66.3 ± 18.1	0.156
	no	25	60.4 ± 21.7	
Cough	yes	48	64.8 ± 17.1	0.829
	no	97	65.5 ± 19.7	
Blocked nose	yes	11	72.7 ± 13.3	0.173
	no	134	64.7 ± 19.1	

**Table 4 jcm-12-07683-t004:** Logistic regression analysis for the prediction of snoring in children aged 3–5 years.

Characteristic	*p* Value	OR	95% CI
Model including breastfeeding with a 3-month cut-off point			
Sleeping with an open mouth, yes	**0.001**	**2.57**	**1.46–4.51**
Adenoid mucus coverage (MASNA scale), 1–3	0.197	1.83	0.73–4.60
Adenoid size, A/C ratio ≥ 75%	**0.023**	**3.59**	**1.19–10.83**
Breastfeeding, <3 months	**0.032**	**4.33**	**1.14–16.50**
Model including breastfeeding with a 6-month cut-off point			
Sleeping with an open mouth, yes	**<0.001**	**2.71**	**1.52–4.84**
Adenoid mucus coverage (MASNA scale), 1–3	0.240	1.73	0.69–4.29
Adenoid size, A/C ratio ≥ 75%	**0.030**	**3.37**	**1.13–10.09**
Breastfeeding, <6 months	0.055	2.66	0.98–7.24

## Data Availability

Additional data supporting reported results may be available for request.

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
