# Peer review of "Impact of Breastfeeding Duration on Adenoid Hypertrophy, Snoring and Acute Otitis Media: A Case-Control Study in Preschool Children"

_jcm, 2023, doi:10.3390/jcm12247683_

Round 1

Reviewer 1 Report

Comments and Suggestions for Authors

The author grouped the children into 6 groups. But the majority of the comparison was not done using these groups

Author Response

Dear Reviewer,

Thank you very much for reading the manuscript, constructive tips and comments, and positive feedback about the work. The answer for questions is attached bellow. Added or changed text is highlighted in green and red

Thank you for your important point, such an initial division of patients was actually based on 6 groups, but in order to determine the significance of the length of breastfeeding in the study group, it was divided into subgroups with a cut-off point of 1,3,6,12, and 24 months. This is supplemented and described in the material and methods. - line 79

Thank You

Reviewer 2 Report

Comments and Suggestions for Authors

This study assessed the relationship between breastfeeding duration and adenoid hypertrophy, snoring and acute otitis media. This paper was well written and well described. I have only two concerns:

-Did the authors consider type As tympanogram as abnormal? It is fairly common to have otitis media with this type of tympanogram among children

-Some statistical results are on the verge of significance (p=0.054). Perhaps authors should consider to use effect size analysis to support the existing statistical outcomes.

Thank you.

Author Response

Dear Reviewer,

Thank you very much for reading the manuscript, constructive tips and comments, and positive feedback about the work. The answer for questions is attached bellow. Added or changed text is highlighted in green and red

1. I agree that in the case of very young children suffered from middle ear effusion , it is possible to get a type As tympanogram -shallow pick < 0.3 mmho, but the study group included older children from 3 years of age in whom obtained type A tympanogram provides very strong evidence that there is no middle ear effusion. It is confirmed in the studies: https://doi.org/10.1046/j.1365-2273.1997.00023.x In our research, tympanogram A indicates a positive pick of at least 0.3 mmho. Besides, type As tympanogram is more characteristic for otosclerosis or tympanosclerosis which affect the elderly.

2.

Yes, we explain it in the line 252. The value you provided refers to the relationship between the 6-category of tympanograms and the acute otitis media. In this case, simplifying the analysis to 3 categories taking into account the worse result of tympanometry allowed to show a significant relationship.  This suggests that in the case of a larger group, also for the 6- categorical variable, a relationship should be demonstrated.

Thank you

Reviewer 3 Report

Comments and Suggestions for Authors

Manuscript ID: JCM-2724820

Reviewer Report

Thank you for giving me a chance to review this study. It is a good quantitative cross-sectional study that present a Impact of breastfeeding duration on adenoid hypertrophy, snoring and acute otitis 2 media. A case-control study in preschool children

Dear author,

This manuscript is well designed and well-presented and it is interesting study in current society. A breastfeeding duration of 1 month or more plays a key role in reducing the rate of  AOM. The mother’s milk plays a protective role against AOM. The presence of mucus might be responsible for snoring in preschool children. A medical history of breastfeeding should be taken into consideration when snoring children are suspected of adenoid hypertrophy. There are some corrections that the authors need to address before the manuscript can be considered for publication.

Comments

1.     Abstract: Kindly add study design, Mention the year of study and duration conducted.

2.     INTRODUCTION: Very less content. Please include more information by referring various studies.

3.     Kindly mention the aim of the study clearly at the end of the introduction stating aim or purpose of the study.

4.     Methods: Authors did not mention the year of study and duration conducted.

5.     Discussion section does not cover some previous studies related to this topic. More studies need to be included and compared to this study results.

1. The obtained results were reiterated in the discussion, and only the results of other studies were mentioned, along with their similarities and differences. Cause and effect relationship were not addressed.

6.     Please highlight how this study adds to the current available knowledge.

7.     Please include future recommendations for this study.

8.     Authors did not mention the Strength of the study…

Best Wishes

Comments on the Quality of English Language

Minor editing of english  language required

Author Response

Dear Reviewer,

Thank you very much for reading the manuscript, constructive tips and comments, and positive feedback about the work. The answer for questions is attached bellow. Added or changed text is highlighted in green and red

Comments

  1. Abstract: Kindly add study design, Mention the year of study and duration conducted. done
  2. INTRODUCTION: Very less content. Please include more information by referring various studies. The introduction was expanded, we included more studies and references in introduction section.
  3. Kindly mention the aim of the study clearly at the end of the introduction stating aim or purpose of the study. Done in line 67
  4. Methods: Authors did not mention the year of study and duration conducted. Done- line 75
  5. Discussion section does not cover some previous studies related to this topic. More studies need to be included and compared to this study results. 
  6. The obtained results were reiterated in the discussion, and only the results of other studies were mentioned, along with their similarities and differences. Cause and effect relationship were not addressed.

The discussion was expanded, the reference list was increased. We including strength of the study and future recommendations– line 249

  1. Please highlight how this study adds to the current available knowledge.- It’s written in line 245
  2. Please include future recommendations for this study. corrected in line 250
  3. Authors did not mention the Strength of the study… line 249

Reviewer 4 Report

Comments and Suggestions for Authors

In their study, the authors investigated the relationship between breastfeeding and some parameters such as adenoids, snoring and acute otitis media in preschool children. The article is overall well organized and written.

Abstract:ok

Introduction: In this section, the formation and effects of adenoid hypertrophy can be briefly mentioned.

Method: The use of soft tissue dose lateral nasopharynx radiography to determine the size of adenoid vegetation can also provide valuable information. The advantages of the procedure performed with a fiberoptic endoscope over direct radiography can be mentioned. Or it should be stated why direct radiography was not used.

Discussion:ok

Kind regards

Author Response

Dear Reviewer,

Thank you very much for reading the manuscript, constructive tips and comments, and positive feedback about the work. The answer for questions is attached bellow. Added or changed text is highlighted in green and red

We agree that lateral nasopharynx radiography may be used to assess the adenoid size. But nowadays flexible endoscopic examination is a gold standard which not only gives the information about adenoid size but also about adenoid mucous coverage which we have also analyzed in the study. The usage of flexible nasopharyngoscopy in adenoid size assessment was analyzed in our work: Zwierz A, Domagalski K, Masna K, Burduk P. Effectiveness of Evaluation of Adenoid Hypertrophy in Children by Flexible Nasopharyngoscopy Examination (FNE), Proposed Schema of Frequency of Examination: Cohort Study. Diagnostics (Basel). 2022 Jul 17;12(7):1734. doi: 10.3390/diagnostics12071734. PMID: 35885638; This study showed that flexible endoscopic adenoid assessment is related to real adenoid size evaluated during its surgical removal. In the study 97.3% sensitivity and 72.7% specificity of flexible endoscopic examination was obtained in the assessment of the adenoid size. Comparing dynamic flexible videoendoscopy to the lateral x-rays which also should be performed at the end of the inspiration, but in that case, it is really difficult to take the single picture in the right moment when we examine frightened and non-cooperating child. It’s true that lateral cephalograms reached 61 to 75% sensitivity and 41 to 96% specificity but systematic review performed by Major suggest that lateral x-rays overestimated the size of the adenoid and should be used rather for the measurement of the size of the airway than the adenoid size. This technique is static and produced two dimensions summation picture. Comparative study performed by Mlynarek showed that in contrast to FNE, lateral x rays’ measurements such as adenoid thickness or A/C ratio did not corelate with obstructive symptom score.  That’s why we chose the flexible endoscopy to asses adenoid size.